# *Chlamydia trachomatis*—An Emerging Old Entity?

**DOI:** 10.3390/microorganisms11051283

**Published:** 2023-05-14

**Authors:** Bogna Grygiel-Górniak, Barbara Anna Folga

**Affiliations:** Department of Rheumatology, Rehabilitation and Internal Diseases, Poznan University of Medical Sciences, 61-701 Poznań, Poland

**Keywords:** *Chlamydia trachomatis*, clinical symptoms, genital infections, trachoma, lymphogranuloma venereum, diagnosis, treatment

## Abstract

*Chlamydia trachomatis* is an evasive pathogen that can prompt severe clinical manifestations in humans such as vaginitis, epididymitis, lymphogranuloma venereum, trachoma, conjunctivitis and pneumonia. If left untreated, chronic infections with *C. trachomatis* can give rise to long-lasting and even permanent sequelae. To shed some light on its widespread nature, data from original research, systematic reviews and meta-analyses from three databases was collected and analyzed in the context of chlamydial infection, related symptoms and appropriate treatment modalities. This review describes the bacterium’s pervasiveness on a global scale, especially in developing countries, and suggests ways to halt its transmission and spread. Infections with *C. trachomatis* often go unnoticed, as many individuals are asymptomatic and unaware of their diagnosis, contributing to a delay in diagnosis and treatment. The high prevalence of chlamydial infection highlights the need for a universal screening and detection method enabling immediate treatment at its onset. Prognosis is favorable with antibiotic therapy and education for high-risk groups and their sexual partners. In the future, a quick, easily accessible, and inexpensive test should be developed to diagnose and treat infected individuals early on. Along with a vaccine against *C. trachomatis*, it would halt the transmission and spread of the pathogen worldwide.

## 1. Introduction

*Chlamydia* is a major bacterial pathogen that infects humans and a wide range of animals, including marsupials, birds, cats, pigs, cattle and sheep [1]. In humans, *C. trachomatis* is an obligate intracellular pathogen responsible most notably for causing sexually transmitted infections (STIs) and is considered the most common cause of curable STIs worldwide [2]. *C. trachomatis* genital infections are a global health concern that cause substantial morbidity, especially in women [3]. Oftentimes, infected individuals are asymptomatic, but pelvic pain, vaginal discharge, urethral discharge and lower abdominal pain are noted in those with symptoms. If left untreated, severe complications can develop, such as pelvic inflammatory disease (PID) and perihepatitis in women and reactive arthritis in men. In addition to these clinical manifestations, *C. trachomatis* precipitates the formation of trachoma, mostly in those residing in developing countries, and is a leading cause of conjunctivitis and pneumonia in neonates [4].

*C. trachomatis* credits its infectivity to a whole host of virulence factors, enabling it to invade and replicate within host cells. These include its cell wall which inhibits phagolysosomal fusion in phagocytes, a type III secretion system (T3SS) which facilitates the entry of pathogenic proteins into the host cell directly [5], and chlamydia protease-like activity factor (CPAF), a serine protease responsible for cleaving host cell proteins [6]. Furthermore, the pathomechanism by which *C. trachomatis* infects a host is unique in that the organism passes through two developmental forms [7], each with its own form and function, with the ultimate goal of gaining entry into the host cell, disrupting its defenses and overwhelming its machinery.

Diagnosis of chlamydial infection is often delayed; this is related to a small sample size or short follow-up time, lack of information on sexual risk behavior or lifestyle factors and misclassification of chlamydia status, as it is primarily based on incidence (nucleic acid amplification tests (NAATs)) [8]. Screening young adults who are sexually active for genital *C. trachomatis* infections is promoted in many high-income countries all over the world, but its effectiveness at the population level is debated [9]. Because of these challenges in clinical practice, it is imperative to possess knowledge regarding epidemiological status, clinical symptoms, complications and therapeutic methods in order to treat this infection and prevent serious complications effectively. This manuscript discusses all of these aspects in the context of more effective screening, available standard antimicrobial therapies and alternative treatments, along with methods aimed at preventing the transmission and spread of *C. trachomatis* infections.

## 2. Materials and Methods

PubMed, Scholar and Cochrane databases of systematic reviews were analyzed from database inception to 31 December 2022. Our search strategy consisted of search strings composed of terms targeting five major areas: (1) epidemiology, (2) transmission pathways (including possible routes and pathomechanism), (3) clinical symptoms (in various gender and age groups), (4) diagnostic methods and (5) treatment and prevention. The combination of the following groups of each area mentioned above with *C. trachomatis* was used. We searched through titles, abstracts and medical subject headings (MeSH). Non-original research articles (e.g., commentaries, editorials, case reports) and studies written in languages other than English were excluded. Original research, systematic reviews and meta-analyses were selected, which reported on community-based or hospital research studies regarding chlamydial aspects.

## 3. Results

An extensive literature review was conducted to learn more about the epidemiology of *C. trachomatis* and its clinical manifestations, complications, pathomechanism and key virulence factors. This was performed to shed light on the pervasiveness of the organism around the globe and to pinpoint strategies to stop its transmission and spread in the human population.

### 3.1. Epidemiology

*C. trachomatis* is the most common cause of bacterial STIs in the world. Epidemiological and clinical data on chlamydial infection rates are difficult to ascertain due to limitations in screening and reporting of infections. Massive screening measures have not been implemented worldwide (mainly in developing countries), contributing to inaccuracies when estimating and calculating prevalence and incidence rates [10]. Furthermore, asymptomatic individuals unknowingly spread the disease to their sexual partners, and, despite treatment, reinfection is common [11]. It is estimated that about 75% of women and 50% of men are asymptomatic [10]. Under these circumstances, obtaining reliable data regarding infection rates is difficult, and prevalence rates are presumably higher than what is recorded in the literature.

Despite these limitations, epidemiological studies have drawn some formidable conclusions. *C. trachomatis* is the most common sexually transmitted pathogen in high-income countries [9], and nearly 130 million infections were reported worldwide in 2012 [12]. Chlamydia affects 4.0% of women of reproductive age and 2.8% of men [12]. In the Netherlands, *C. trachomatis* prevalence among women between the ages of 16 and 34 reaches 3% [13]. Recent data showed that 10–30% of women experience one or more chlamydia episodes during their lifespan [14,15]. Tubal factor infertility (TFI) develops in approximately 1% of women after chlamydial infection [16]. In populations where *C. trachomatis* is prevalent, TFI results in a considerable number of infertile women. In the Netherlands, TFI is considered a childlessness factor in nearly 11% of infertile couples [17]. In the United Kingdom, the risk of PID (symptomatic or asymptomatic) following an untreated *C. trachomatis* infection is 17.1%, while the risk of salpingitis is 7.3% [18].

*C. trachomatis* is the most common STI etiology among sexually active males 14 to 35 years of age [19]. This pathogen is also the most common cause of nongonococcal urethritis (NGU), accounting for 20% to 40% of NGU cases [20,21]. In males, NGU is characterized by dysuria and pruritus, urethral discharge, elevated polymorphonuclear leukocytes (PMNs) in the urethra, and the absence of *Neisseria gonorrhoeae* [22]. Like many females, many males do not know that they are infected; approximately 50% of *C. trachomatis* infections in men are asymptomatic [23,24]. Nevertheless, severe complications caused by *C. trachomatis* have been reported, such as acute epididymitis observed in men younger than 40 years old (28% of patients) [25].

Infection rates are highest among adolescents and those aged 15–24 [11]. This age group accounts for two-thirds of new infections, with females being predominantly affected [26]. Some reports suggest that one in twenty sexually active young females aged 14–24 has chlamydia [27].

Trachoma is endemic to certain geographic regions in the world, and approximately 1.2 billion people are estimated to reside in trachoma-endemic areas [28]. Active trachoma is detected mainly in Africa, the Mediterranean Region, Southeast Asia, the Western Pacific and the Americas [29]. About 41 million people suffer from active trachoma [28], and the disease process is responsible for blindness or visual impairments in about 1.9 million individuals worldwide [30].

### 3.2. Chlamydial Biovars and Serovars

Symptoms related to chlamydial activity largely depend on specific biovars and serovars. There are three main human biovars—trachoma, genital tract, and lymphogranuloma venereum (LGV)—characterized by specific clinical symptoms (Figure 1). Immunological data and protection implications identified 15 major *C. trachomatis* serovars [31,32]. The trachoma biovar is subdivided into serovars A–C, leading to trachoma, the most common cause of infectious blindness in developing countries [7]. The genital tract biovar splits into serovars D–K, responsible for genital tract infections and severe complications in both sexes if left untreated. Women can experience cervicitis, urethritis, PID, or perihepatitis [4]. Infertility and ectopic pregnancy represent long-lasting sequelae of infection [2]. In men, urethritis, epididymitis, prostatitis, proctitis, or reactive arthritis may develop [4]. Serovars D–K are also implicated in neonatal infections, of which ophthalmia neonatorum (conjunctivitis) and pneumonia constitute the most common clinical findings [33]. The last biovar, LGV, is sectioned into serovars L1–L3, which cause invasive urogenital or anorectal infections [7,34]. These serovars also contribute to the development of genital ulcers. Other complications of LGV include lymphadenopathy and fibrosis [35].

### 3.3. Modes of Transmission/Pathomechanism

*C. trachomatis* can be transmitted through a variety of means. Infection can occur after oral, anal, or vaginal sex, but can also be passed on from mother to offspring during childbirth [33]. Trachoma can be acquired through person-to-person contact with infected ocular and nasal passages, through fomites, or via eye-seeking flies [28]. The manner in which *Chlamydia* infects and replicates within a host cell is distinct (Figure 2). The bacterium assumes two developmental forms, the elementary body (EB) and the reticulate body (RB) [4]. The EB, which is metabolically inactive and infectious, enters the host mucosal cell, binding with the help of a trimolecular bridge formed between bacterial adhesions, host cell receptors, and host heparan sulfate proteoglycans (HSPGs). From there, pre-synthesized T3SS effectors are infused inside the host cell, promoting the EB’s internalization process and formation of an anti-apoptotic state. The EB is introduced into the host cell via endocytosis and is stored within a membrane-bound compartment called an inclusion. Inside the inclusion, bacterial proteins are synthesized and the EB transforms into an RB, which is metabolically active and non-infectious. Newly synthesized inclusion membrane proteins (Incs) aid in the process of obtaining nutrients by diverting vesicles secreted from the Golgi apparatus meant for the plasma membrane to the inclusion [7,36]. As a result, Incs can promote fusion with host-cell nutrient-rich compartments, inhibit fusion with degradative organelles, and take over host cell machinery to disrupt pathways native to the host, generating new complexes with functions benefiting the pathogen [37]. The inclusion is then transported to the centrosome [7,36]. There, the RB can replicate at an alarming rate, depleting the host’s energy supply in the process [4]. Following several rounds of replication, the RB retransforms back into an EB, which is released from the host cell, capable of entering more cells and carrying on the infective and replicative cycles [7].

### 3.4. Virulence Factors

*C. trachomatis* possesses virulence factors which contribute to its pathogenicity and infectivity. These factors include various surface antigens [5,38] type III and type IV secretion systems [39,40,41], chlamydial plasmids [42], and genetic variations, which impact variability and expression of virulence factors [34,43,44]. Its atypical developmental cycle and intracellular life forms [41,45,46], along with its ability to stimulate apoptosis [45], contribute to the virulence of this pathogen.

The surface antigens include outer membrane proteins, lipopolysaccharide, heat shock proteins and polymorphic membrane proteins (Pmps) responsible for the essential first step in infection [5,38]. In addition to these surface antigens, the chlamydial cell wall has the ability to inhibit phagolysosome fusion in phagocytes, promoting survival in the host cell.

Furthermore, the T3SS is crucial for the bacterium’s survival and virulence. It is a non-flagellar system acting like a “molecular syringe,” delivering anti-host bacterial “effector” proteins directly into a host cell in a contiguous process [47,48]. These injected proteins have been known to promote the bacterium’s viability in the host cell by blocking host cell signaling pathways and overwhelming host cell machinery [49]. T3SS effectors may function as structural determinants of the membrane or as scaffolds to interface with various cell pathways in the host [50]. The T3SS allows *C. trachomatis* protein products to enter the host cell’s cytoplasm directly, bypassing host lysosomes and assisting in invasion and replication [5]. In addition to the stimulation of effector proteins [50,51], T3SS also boasts immunostimulatory capability [52]. T3SS effectors are regulated by CPAF (chlamydial protease-like activity factor). CPAF is also responsible for cleaving host cell proteins, maintenance of inclusion membrane proteins (Incs), and evasion of caspase-1-dependent cell death in epithelial cells, consequently establishing an anti-apoptotic state in cells targeted by *C. trachomatis* [53].

The virulence factors belonging to *C. trachomatis* have been shown to induce various mechanisms which enable the bacterium to manipulate host functions. These include surface antigens responsible for stable adhesion to host cells and possibly host cell immune escape, membrane proteins, lipopolysaccharide, heat shock proteins and Pmps [5,38]. Pmps are monomeric proteins that can function as toxins, proteases, lipases, or mono-adhesive adhesins [54]; they are localized on the surface of the chlamydial cell, where most of them are proteolytically processed [38]. They constitute the largest chlamydial protein family, with 9 members (subdivided into six subtypes) in *C. trachomatis* [55].

Another virulence factor is chlamydial cytotoxin for epithelial cells, such as putative large cytotoxin. It is transiently present in infected cells during the period of cytotoxicity. Putative large cytotoxins are detected in the EB and are delivered to host cells early on in infection. Their cytotoxic activity causes morphological and cytoskeletal changes in epithelial cells that are indistinguishable from those mediated by clostridial toxin B [56].

Crucial for *C. trachomatis* virulence are chlamydial plasmids [57], which contain both noncoding RNAs and eight open reading frames (ORFs) [57,58,59]. All ORFs are conventionally called plasmid glycoproteins 1–8 or pGP1–8 [60] and are expressed in infected cells [61,62]. In murine models, these plasmids can enhance proinflammatory cytokine stimulation through the involvement of Toll-like receptors (TLR) [63]. In vivo, infections with plasmid-deficient organisms either are asymptomatic or exhibit significantly reduced pathology [64,65]. In animal models, plasmid-deficient *C. trachomatis* do not cause trachoma but induce strong protective immunity to fully virulent plasmid-bearing bacteria. Thus, plasmid-deficient organisms can be used to lay the foundation for the novel live-attenuated chlamydial vaccine [64].

Several types of genetic variation are found in *C. trachomatis* that impact the variability and expression of virulence factors. Recent data suggest that a few genetic features are involved in phenotypic dissimilarities in *C. trachomatis* infections. They include recombination and point mutations of Incs and T3SS effectors [34]. These genetic variations promote chlamydial intracellular survival and influence disease severity. For example, the *Tarp* gene variation (locus ct456) alters the number of actin-binding domains and the internalization rate of the pathogen [43]. Subtle variations in the amino acids of a subset of Inc proteins and the expression of Inc genes contribute to the unique tropism and invasiveness of *C. trachomatis* LGV strains [44].

### 3.5. Clinical Signs and Symptoms

Many patients with chlamydia remain asymptomatic, but a minority of individuals develop symptoms that depend on the location of infection [4]. Infected females, males, and neonates can present with characteristic findings.

#### 3.5.1. Females

In females, the cervix is the site most often colonized by *C. trachomatis* [4]. Consequently, cervicitis can occur, and women may experience mild symptoms such as vaginal discharge, bleeding, abdominal pain and dysuria [66]. Some women may present with mucopurulent cervicitis, endocervical bleeding and postcoital or intermenstrual bleeding. If the infection ascends from the cervix into the upper reproductive tract, patients experience abdominal or pelvic pain, and the infection advances to PID [4]. Apart from the pain, other symptoms have been reported, including nausea, vomiting, fever, chills, low back pain, dysuria, dyspareunia or pain during sexual intercourse, and postcoital bleeding [67]. Patients diagnosed with PID may develop Fitz–Hugh–Curtis syndrome (FHCS), or perihepatitis, a condition in which the liver and surrounding peritoneal surfaces become inflamed, prompting right upper quadrant (RUQ) or pleuritic pain [4]. FHCS is frequently associated with symptoms of PID (fever, lower abdominal pain, vaginal discharge) [68]; however, it can be complicated by long-term sequelae such as infertility, ectopic pregnancy and chronic pelvic pain [69]. In addition to this, PID (particularly if untreated) may lead to tubal scarring due to an intense and chronic inflammatory response [70]. Tubal scarring may result in TFI [71].

Other complications of untreated or long-standing chlamydial infection in the reproductive tract, particularly in the fallopian tubes, can lead to infertility and ectopic pregnancy [72]. In pregnant women, chlamydial infection has been associated with adverse pregnancy outcomes such as premature rupture of membranes (PROM), preterm birth, low birth weight, growth restriction and neonatal death [73]. While uncommon, females infected with *C. trachomatis* can experience urethritis, where urinary frequency and dysuria are the chief complaints [4].

#### 3.5.2. Males

Symptomatic males may exhibit a combination of urogenital and extragenital manifestations. Urogenital infection in males can present with epididymitis, in which males note unilateral testicular pain and tenderness with associated swelling of the epididymis. Urethritis is also a common sign of chlamydial infection in males; patients experience dysuria and often observe a white, gray urethral discharge. Prostatitis is evidenced by dysuria, pelvic pain, urinary dysfunction and dysorgasmia [4].

Extragenital findings include proctitis and reactive arthritis. Proctitis, or rectal inflammation, is painful when caused by serovars L1–L3, and patients suffer from rectal discharge, bleeding, fever and malaise. This condition is almost exclusively limited to males who have sex with males [4]. Reactive arthritis is another established manifestation of chlamydial infection. It is estimated that 1% of males with urethritis caused by *C. trachomatis* develop reactive arthritis [74]. On physical examination, asymmetric oligoarthritis, usually of the lower extremities, and sausage-shaped finger, toe, or heel pain, are demonstrated. Patients may also develop a triad of arthritis, urethritis, and uveitis, a condition formerly known as Reiter syndrome [75].

#### 3.5.3. Males and Females

##### Lymphogranuloma Venereum

LGV and trachoma are clinical manifestations of chlamydial infection that are spotted in both sexes. LGV is classified as an ulcerative disease of the genital region [76]. The disease follows a three-stage course [77,78]. The formation of painless genital ulcers marks the first or primary stage. These often go unnoticed due to their size and location and may heal spontaneously. The development of tender inguinal and/or femoral lymphadenopathy usually follows and sets the secondary stage of infection in motion. During this stage, patients may experience a proctocolitis-like illness defined by dysuria, dyschezia or difficulty passing stool, abdominal, rectal, and anal pain, and tenesmus, or a frequent urge to pass stool. Constitutional symptoms such as fever, headaches and body aches have been described. The final late stage of infection appears when the disease is left untreated and is characterized by strictures, fibrosis, fistula formation in the anogenital area, and necrosis in and rupture of the affected lymph nodes [78]. Overall, males present early on in the disease course when acute symptoms are present, while females present during the later stages of infection when complications arise [79].

##### Trachoma

Trachoma is a leading cause of blindness in the world today. Common features upon presentation include redness, itching and irritation of the eyes and eyelids [80]. Discharge, swelling, pain and photophobia are also exhibited. Two forms of trachoma have been described: the active form and the chronic cicatricial form. Active trachoma is more common in children and is characterized by mixed follicular and papillary conjunctivitis. The condition is associated with mucopurulent discharge, superior epithelial keratitis and corneal vascularization. In severe cases, papillary hypertrophy may develop. Recurrent infection gives rise to the cicatricial form of trachoma, seen in middle-aged adults. This chronic stage is marked by scarring, particularly in the upper tarsal plate, along with corneal vascularization, trichiasis and distichiasis. Over time, the destruction of goblet cells in the conjunctiva and ductules of the lacrimal glands precipitates dryness in the eyes. This process and the development of corneal opacification and blindness are hallmarks of permanent damage [81].

#### 3.5.4. Neonates

Neonates may become infected with *C. trachomatis* from their mothers when passing through the vaginal canal during childbirth. Interactions with infected genital secretions can result in conjunctivitis, or ophthalmia neonatorum, in the newborn, characterized by erythema and edema of the eyelids, palpebral conjunctivae and conjunctival discharge [82]. Conjunctivitis is considered the most common clinical manifestation of chlamydia in newborns [4]. Symptoms typically appear in one eye 5–14 days after delivery and the second eye usually becomes inflamed following another 2–7 days [83,84]. The discharge is watery at the beginning but then becomes purulent after a few days [85]. If left untreated, reports of corneal and conjunctival scarring have been documented [86].

Aspiration of infected genital secretions during childbirth may result in pneumonia in the newborn [84]. Symptoms appear between 4 and 12 weeks of age. Patients are usually afebrile, produce a paroxysmal staccato cough, and may have tachypnea [4]. In addition to these clinical signs, nasal congestion and thick nasal secretions are common [87]. A chest X-ray will reveal diffuse pulmonary infiltrates along with hyperinflation [85]. Rales may also be heard on auscultation during physical examination [4]. If left untreated, infants are more susceptible to developing chronic pulmonary disease, including asthma [85].

### 3.6. Clinical Complications

Each diagnosed infection should be treated because a lack of therapeutic management may result in disease progression and complications characterized by long-lasting sequelae [Table 1].

The most severe complications are related to ineffective management and treatment of chlamydial infections. For example, the prevalence of PID following chlamydial infection was found to be between 3.0% and 30.0% [18,94,95]. Ectopic pregnancy was also reported in 0.2% to 2.7% of infected women [18,96,97], while TFI was detected in 0.1% to 6.0% of infected women [18,98].

### 3.7. Screening for Chlamydia

The United States Preventive Services Task Force (USPSTF) recommends that all sexually active females 24 years or younger be screened for chlamydia annually. Similar protocols should be implemented in females 25 years old and over who are at increased risk of infection (new sexual partner, more than one sexual partner, a sexual partner with multiple concurrent sexual partners, or a sexual partner with a diagnosed STD) [99]. In addition, pregnant women under the age of 25 and those over 25 who have an increased risk of infection (new sexual partner, more than one sexual partner, a sexual partner with multiple concurrent sexual partners, or a sexual partner with a diagnosed STD) should be screened at their first prenatal visit. Those at an increased risk at that time should be retested during the third trimester [100].

The Centers for Disease Control and Prevention (CDC) also recommends screening certain male populations, mainly sexually active males in high-prevalence areas and populations. Annual screening should be performed in males who have sex with males; however, more frequent testing is advised in groups who engage in high-risk sexual behaviors or if sexual partners are found to have multiple other partners. Transgender individuals should be screened based on their sexual participation and respective anatomies. Moreover, the CDC recommends screening for females under 35 years of age and males under 30 years of age at the time of entering a correctional facility [100].

In all of the aforementioned situations, patients should likewise be screened for gonorrhea, which often coexists with chlamydia [99]. In cases of a chlamydial or a gonococcal infection, the patient ought to be retested three months after the initiation of treatment (regardless of the infective status of their sexual partner/partners) [100].

### 3.8. Diagnostics/Sampling

Nucleic acid amplification tests (NAATs) are deemed the most sensitive for detecting chlamydia [8]. They have since replaced culture as the diagnostic gold standard of choice. Antigen tests, such as enzyme immunoassays, direct fluorescent antibody (DFA) assays, and rapid diagnostic tests, which were once utilized for detection and diagnosis, are also no longer recommended because of inaccuracies in the diagnostic workup procedure [101,102]. There are a multitude of reasons for this exchange. First, because NAATs do not rely on infectious or viable pathogens, specimens can be easily collected and transported [8]. Furthermore, non-invasive samples such as urine can be analyzed, screening for infections in asymptomatic individuals who would otherwise go undiagnosed and untreated. This proves advantageous since the majority of chlamydial infections in the female population and a cohort in the male population are asymptomatic [2]. Lastly, because most of the NAATs rely on polymerase chain reaction (PCR) and fluorescently labeled probes to locate amplified sequences in real-time, the testing process is more efficient, and the test duration time is reduced. Results are thus ascertained quickly, often within a few hours [8].

In order to diagnose genital chlamydia using a NAAT, vaginal swabs are collected in females and urine is collected in males (Figure 3). For rectal or pharyngeal chlamydial infections, testing at the exposure site is warranted [33]. Chlamydial ophthalmia in neonates should be diagnosed by both tissue culture and nonculture tests (e.g. DFA tests and NAATs). Both methods are sensitive and specific [100]. DFA is the only nonculture FDA-cleared test for detecting chlamydia from conjunctival swabs (NAATs are not cleared by FDA for detecting chlamydia from conjunctival swabs). For culture and DFA, specimens must contain conjunctival cells (obtained from the everted eyelid), not exudate alone [82,100].

In addition to using NAATs and DFA tests, chlamydia-specific antibodies can be detected in the patient’s serum. The two major antibodies in genital secretions that may prevent *C. trachomatis* transmission are IgG against *C. trachomatis*, the predominant isotype, and polymeric secretory IgA [103,104]. Both antibodies are mainly synthesized by local plasma cells in the upper female genital tract (FGT) [105]. Both of these antibody isotypes prevent infection caused by *C. trachomatis* by blocking the entry of bacteria into the host cell. They entrap the bacteria in mucus in the lumen of the FGT [106] or neutralize intracellular pathogens within columnar epithelial cells during transport [107,108]. Antibodies enhance chlamydial opsonophagocytosis and degradation, curbing the infective process (which would have otherwise relied on interferon-gamma (IFN-γ) synthesis) [109]. Plasma cells that synthesize IgG against *C. trachomatis* are found in the FGT, but genital IgG is mainly derived from circulation [110]. In contrast, subepithelial plasma cells produce genital IgM and IgA [111,112], and nearly 70% of the IgA is locally produced in women [113].

IgA and IgM classes are known to react quickly to acute reinfections. The main location of the production of secretory IgA is in the cervix [114]. Similarly, secretory IgA is the predominant isotype secreted by the intestinal mucosa, but in the FGT, a greater IgG to secretory IgA ratio has been observed [115,116]. Human longitudinal studies showed that chlamydia-specific CD4 T-cell IFN-γ responses (but not IgG titers) can reduce the risk of reinfection in highly exposed women [117,118]. The high IgG titers are a marker of repeated and/or prolonged chlamydial exposure, but they do not protect from reinfection [119,120]. Serum titers of anti-EB IgG are associated with reduced cervical burden and an overall decrease in the risk of endometrial infection [117]. The role of anti-IgG antibodies is inconclusive. Some data suggest that anti-EB IgG is ineffective at limiting endometrial ascension [121]. In contrast, others report that anti-chlamydia IgA (but not IgG) might limit ascension and prevent the formation of salpingitis [122].

### 3.9. Treatment

Overall, treatment is indicated to suppress the development of complications associated with chlamydial infection, reduce the risk of transmission of the organism, and aid in the resolution of ongoing symptoms [4]. The antibiotic regimen administered depends on the severity of the infection, the types of symptoms, and the patient’s age [Table 2].

#### 3.9.1. Females

Uncomplicated genital chlamydial infections in females are most commonly treated with a one-time oral dose of 1 g azithromycin. Oral doxycycline at a dose of 100 mg twice daily for 7 days may be substituted [4]. Patients should abstain from sexual activity until they complete treatment [33]. For complicated cases which have progressed to PID, the conventional parenteral antibiotic regimen consists of ceftriaxone 1 g IV every 24 h, doxycycline 100 mg orally or IV every 12 h and metronidazole 500 mg orally or IV every 12 h. The transition to oral therapy can usually begin within 24–48 h of clinical improvement of symptoms. Treatment for PID should include coverage for *Neisseria gonorrhoeae* [100]. Furthermore, adding metronidazole to the PID regimen protects against anaerobic organisms in the upper genital tract [124]. If perihepatitis is diagnosed, treatment is similar to that of PID, but hospitalization may be required if patients are pregnant or are immunocompromised, if a pelvic abscess is detected on imaging, or if patients fail to improve after 72 h of therapy. Surgical intervention such as laparoscopy may be deemed necessary for symptomatic adhesiolysis and a laparotomy may be performed in cases of surgical emergencies [125].

Pregnant women infected with *C. trachomatis* are advised to take a one-time oral dose of 1 g azithromycin. Alternative treatment regimens include 500 mg oral amoxicillin three times daily for 7 days or 500 mg oral erythromycin twice daily for 7 days. Azithromycin remains the standard treatment of choice [4]. In comparison to erythromycin, azithromycin has been shown to cause significantly fewer gastrointestinal side effects in expecting mothers [132].

Women with TFI planning conception may need medical assistance, for example, in vitro fertilization (IVF), to become pregnant [133]. Unfortunately, successful IVF ending in a live birth occurs in about only approximately 42% after three complete IVF cycles [134].

#### 3.9.2. Males

The treatment of uncomplicated genital chlamydial infections in males is the same as that for females. For anorectal chlamydial infections, often seen in men who have sex with men [MSM], a regimen containing 100 mg oral doxycycline twice daily for 7 days is preferred over a one-time 1 g oral dose of azithromycin [4]. Side effects such as nausea, vomiting, and diarrhea were reported with ingestion of both antibiotics [135].

Treating reactive arthritis in males, on the other hand, has been challenging. In recent years, non-steroidal anti-inflammatory drugs (NSAIDs), antibiotic therapy, and disease-modifying anti-rheumatic drugs (DMARDs) were studied for their ability to manage chlamydia-induced reactive arthritis successfully. The purpose of therapy is twofold: to alleviate symptoms of reactive arthritis while also preventing the development of chronic complications. NSAIDs remain the standard treatment of choice in acute settings, while DMARDs, such as sulphasalazine, are prescribed in both acute and chronic conditions or when NSAIDs are ineffective. Antibiotics are administered for three to six months when an infectious agent, such as *Chlamydia*, is the cause of symptoms. In addition to these therapies, patients are advised to exercise and perform stretching exercises to prevent muscle wasting [75].

#### 3.9.3. Lymphogranuloma venereum

Administration of doxycycline 100 mg orally twice daily for 21 days is the recommended antibiotic regimen for treating LGV. Alternatively, 500 mg of oral erythromycin can be prescribed four times daily for 21 days [131]. In addition to antibiotic therapy, aspiration and drainage may be required to inhibit the formation of ulcers [11]. All patients suspected of having LGV, along with any of their sexual partners, should be treated empirically. This regimen includes either 100 mg oral doxycycline twice daily for 7 days or a one-time dose of 1 g oral azithromycin. Testing for LGV and chlamydial infection should be performed, and if either of those tests returns with a positive result, treatment should be carried out for a total of 21 days. If the results are negative, treatment can be discontinued after 7 days [76].

#### 3.9.4. Trachoma

A single dose of oral azithromycin (starting at 20 mg/kg up to 1 g) is the standard treatment of choice for individuals diagnosed with trachoma as well as for those exposed. Alternatively, 500 mg oral erythromycin can be administered twice daily for 14 days or 100 mg oral doxycycline daily for 10 days. An amount of 1% tetracycline ointment can be applied topically, but oral treatment is preferred. Surgical intervention may be necessary for severe symptoms such as trichiasis [81]. Since trachoma is considered a global syndrome, the World Health Organization (WHO) started the Global Elimination of Trachoma (GET 2020) initiative to eliminate blinding trachoma by 2020. Community-wide control interventions were developed, known as the ‘SAFE’ strategy: Surgery to treat those suffering from trachomatous trichiasis, massive drug administration (MDA) with Antibiotics, promotion of Facial cleanliness, and Environmental improvement to reduce prevalence and transmission of the causative agent [136]. After the implementation of the SAFE strategy, the population prevalence of trachomatous trichiasis and trachomatous follicular inflammation in 1 to 9-year-olds decreased to <0.1% and below 5%, respectively [80,137].

#### 3.9.5. Neonatal Infections

Treatment of ophthalmia neonatorum consists of either a regimen of erythromycin 50 mg/kg/day for 14 days or azithromycin 20 mg/kg/day for 3 days [127]. Erythromycin is also used to manage chlamydial pneumonia in infants [85]. A second antibiotic course is usually prescribed because 20% of cases recur; eradicating the organism in this patient population has been challenging. However, complications arose when treating chlamydial infection in such young patients. Pyloric stenosis was observed in infants less than 6 weeks old who were previously treated with erythromycin [127]. Patients should consequently be monitored for any signs of intestinal obstruction [4].

#### 3.9.6. Antibiotic Resistance

Multidrug resistance (MDR) has increased worldwide and is considered a public health threat. Several recent investigations have reported the emergence of multidrug-resistant bacterial pathogens from different origins, showcasing the necessity for proper use of antibiotics. There have been a few documented reports of antibiotic resistance in *C. trachomatis*; however, recent data point to some possible mechanisms responsible for bacterial growth during antibiotic exposure [138]. These are outlined in Table 3.

Multiple antibiotic-resistance genes can be readily recombined between *Chlamydia* spp. The possibility of recombinant transfer of TET resistance from *C. suis* to *C. trachomatis* and *C. muridarum* strains is one of the mechanisms which can participate in antibiotic resistance [141]. Recombination in *Chlamydia* spp. occurs naturally; therefore, clinical resistance might spread rapidly in patients. Another mechanism that can contribute to antibiotic resistance is the transformation via electroporation in chlamydia using SPC and KSM [146]. The *C. suis* TET-resistant strains and the in vitro results with the other antibiotics (described above) create a possibility of developing antibiotic resistance. Fortunately, there is no genetic evidence of antibiotic resistance leading to treatment failure in humans.

Only a few reports describe the isolation of antibiotic-resistant *C. trachomatis* strains from patients, but only a small portion of the population (<1–10%) expressed resistance [150,151]. All described above isolates display ‘heterotypic resistance,’ a form of phenotypic resistance in which a small proportion of an infecting microbial species can express resistance at any time. Heterotypic resistance is not typical for *Chlamydia* spp., but is also reported in the case of *Staphylococcus* spp. infection [152,153].

## 4. Discussion

### 4.1. Prognosis

Since *C. trachomatis* causes acute and chronic chlamydial infections all over the world [4], as well as trachoma, mainly in countries with poor sanitary conditions [154], there is an urgent need for effective protection against the pathogen. Prognosis depends on hygiene, education, socioeconomic status and access to healthcare. Antibiotics are the only treatment currently available. Unfortunately, screening and antibiotics treatment programs have not resulted in a reduction in infection rates. Antibiotic therapy presumably results in a halted natural immune response that would have otherwise facilitated reinfection [155]. However, high rates of reinfection show that there is a need to develop a vaccine against *Chlamydia* [1]. An analysis of the major outer membrane protein (MOMP) in sequences of *C. trachomatis* showed the presence of variable domains (VD), regions of DNA unique to each serovar [156]. The serovar/serocomplex protection elicited during the trachoma vaccine trials was due to MOMP [157]. Therefore, it was hypothesized that a polyvalent vaccine formulated with the senior serovar of each complex would protect against all the individual serovars [158].

### 4.2. Prevention

Certain public health policies have been endorsed in order to prevent infection with and transmission of STDs, such as *Chlamydia*. In addition to these policies, safe sexual practices are encouraged, e.g., primary prevention tactics such as sexual health education [159]. Behavioral counseling interventions adopted by primary care physicians aim to provide individuals with information on STDs, assess patients’ risks for developing an STD, advocate for the use of condoms and discuss safe sexual habits [160]. Monogamy is one such habit [161]. Aside from these steps, screening for asymptomatic infections can halt the transmission and spread of an STD as well [162].

Prevention of reinfection is also a public health priority. Some measures to prevent reinfection in individuals are notifying, testing and treating sexual partners from the last six months [159]. Moreover, sexual partners should abstain from condomless sexual activity until all involved parties are treated accordingly and any symptoms present at the time of disease are resolved [100].

## 5. Conclusions

Chlamydia continues to be a global health concern. Asymptomatic infections and a lack of universal screening and diagnostic procedures have made the organism easy to transmit but difficult to detect. Those most at risk include males who have sex with males, young, sexually active females, pregnant women and neonates. Treatment is usually effective, but reinfection is common. If untreated, individuals may develop a whole host of severe complications, including PID, FHCS and infertility (TFI). Prevention of chlamydial infection is crucial, and education should be provided to those at risk along with their sexual partners. These factors prompt the need for a quick, easily accessible and inexpensive test for those at risk worldwide. In this way, diagnosed individuals can seek treatment early on. Future studies should pilot such a method and map out plans for a vaccine protecting against chlamydia. Combined therapy and prevention would curb the spread and transmission of the pathogen globally.

## Figures and Tables

**Figure 1 microorganisms-11-01283-f001:**
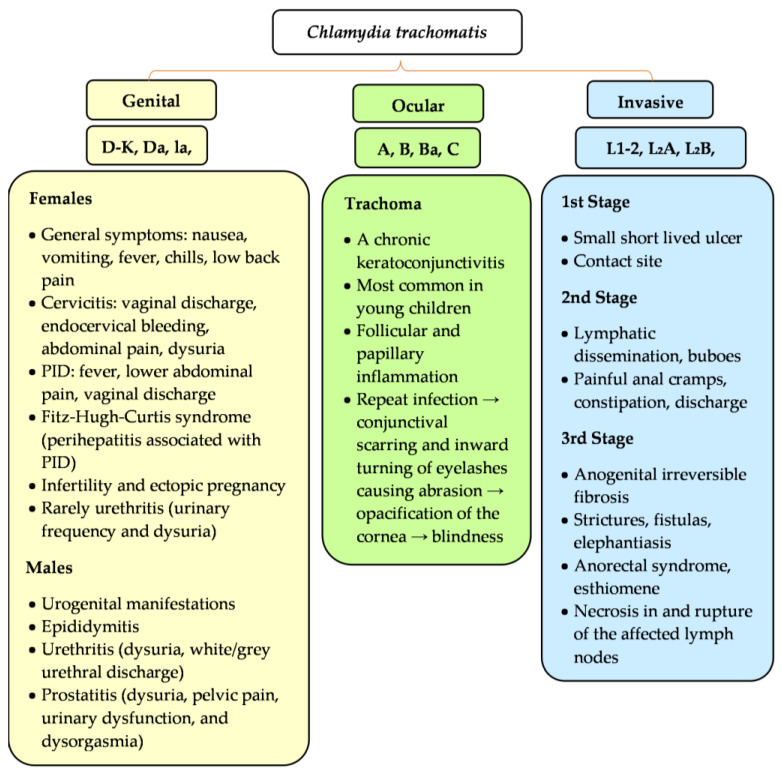
Serological and clinical characteristics of *Chlamydia trachomatis*. Characteristics of *C. trachomatis* serovars and clinical symptoms. Serovars can infect and survive in diverse host niches causing wide spectrum urogenital (serovars D–K, Da, Ia, Ja) and ocular symptoms (serovars A–C). The lymphogranuloma venereum (LGV) serovars (L1–L3) invade macrophages and spread systemically through lymph nodes.

**Figure 2 microorganisms-11-01283-f002:**
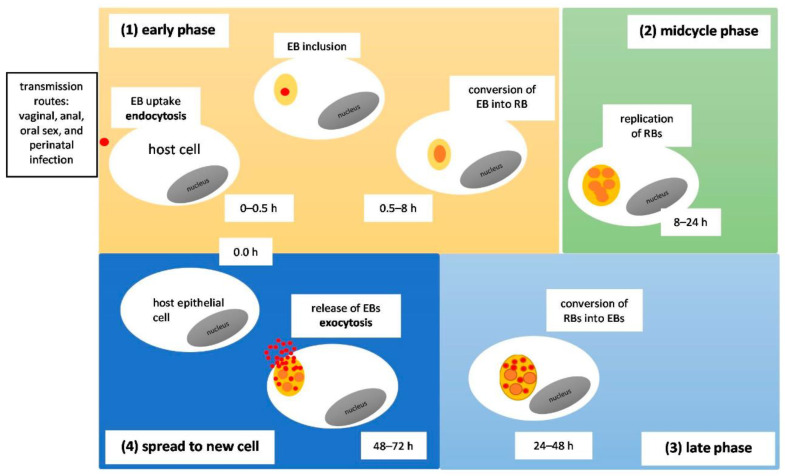
The Chlamydial Developmental Cycle. (**1**) Early phase: Adhesion of *C. trachomatis* elementary bodies (EBs; ∼0.3 µm in diameter) to the epithelial host cell. EBs are the small, infectious and non-replicative form. They adhere to the surface of host cells, triggering the delivery of T3SS effectors and enabling chlamydial invasion. As a result, chlamydial internalization and inclusion (formation of a membrane-bound compartment) is observed and this process lasts about 0 to 2 h post-infection. Then, the EBs differentiate into RBs (RBs; ∼1 µm in diameter) about 2–8 h post-infection. (**2**) Midcycle phase: Intravacuolar RBs begin replicating and cause a large inclusion occupying most of the host cell cytoplasm (∼6–24 h post-infection). (**3**) Late phase: The RBs re-differentiate asynchronously into EBs (∼24–48 h post-infection). (**4**) Spread to new cell: The inclusion is then filled with EBs (the infectious progeny), and a few lasting RBs are released by the host cell lysis or extrusion (∼48–72 h post-infection). From there, the infectious progeny can infect neighboring cells.

**Figure 3 microorganisms-11-01283-f003:**
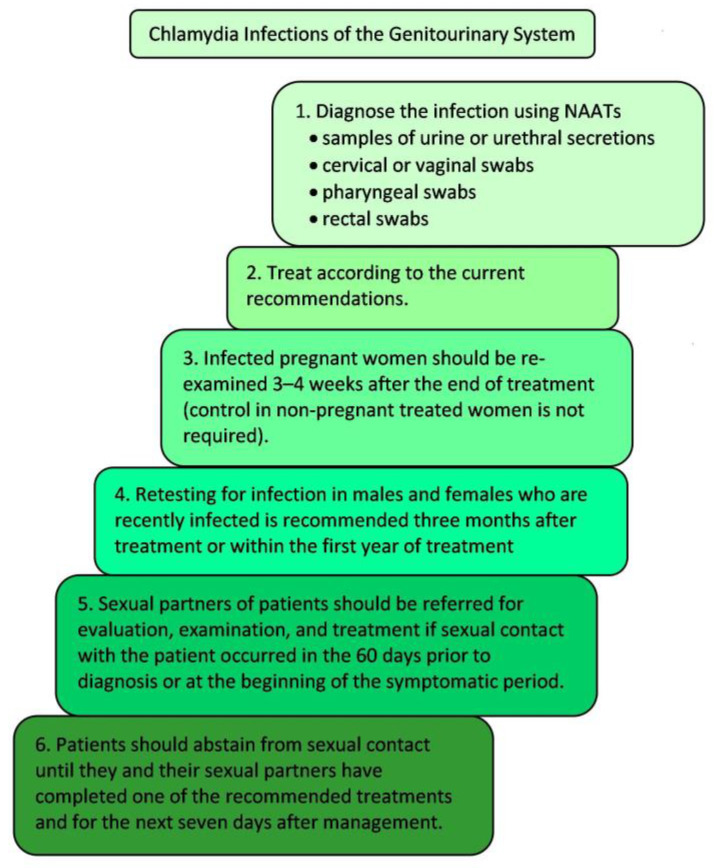
Steps in management of a genitourinary system infection caused by *C. trachomatis*; NAATs—*Nucleic Acid Amplification Tests*.

**Table 1 microorganisms-11-01283-t001:** Clinical complications of *Chlamydia trachomatis* infection.

Risk Group	Possible Complications	References
Females	Pelvic inflammatory disease Fitz-Hugh-Curtis syndrome Chronic pelvic pain syndrome Ectopic pregnancy Infertility (including tubal factor infertility) Probable ↑ risk of HPV co-infection → ↑ risk of cervical cancer	[70,71,88]
Pregnancy	↑ risk of pregnancy complications: Premature rupture of the membranes Premature birth Low birth weight Miscarriage Neonatal death	[89]
Males	Epididymitis Orchitis Infertility	[90]
Females and Males	↑ susceptibility to HIV infection Risk of developing Reiter’s syndrome (aseptic arthritis, nongonococcal urethritis, conjunctivitis)	[91,92,93]

**Table 2 microorganisms-11-01283-t002:** Treatment of various forms of chlamydial infection.

*Chlamydia trachomatis* Infection			
Chlamydia Infection	Type of Infection	Standard Treatment	Alternative Regimens
Genitourinary system [4,100,123,124,125]	Uncomplicated	1 g azithromycin p.o. single dose or Doxycycline 2 × 100 mg p.o. for 7 days; (equally effective as azithromycin) Abstain from sexual activity until recovery	Erythromycin 4 × 500 mg for 7 days (erythromycin causes gastrointestinal adverse effects more often than other alternative drugs) Erythromycin ethyl succinate 4 × 800 mg for 7 days Levofloxacin 1 × 500 mg for 7 days Ofloxacin 2 × 300 mg or 1 × 600 mg for 7 days
	Complicated (e.g., PID or perihepatitis)	Combined therapy ceftriaxone 1 g IV every 24 h or doxycycline 100 mg orally or IV every 12 h with Metronidazole 500 mg orally or IV every 12 h	Symptomatic adhesiolysis requires laparoscopy Severe symptoms—laparotomy
Pregnancy [4,126]		Azithromycin 1 g one-time oral dose (standard treatment of choice) or 500 mg oral amoxicillin three times daily for 7 days	Erythromycin 4 × 500 mg for 7 days or 4 × 250 mg for 14 days Erythromycin ethyl succinate 4 × 800 mg for 7 days or 4 × 400 mg for 14 days (Similar efficacy of all drugs, but azithromycin has fewer side effects than erythromycin or amoxicillin)
Male [4]	Uncomplicated	Genital chlamydial infections—the same as that for females	Same treatment as females
	Complicated	Anorectal chlamydial infections (mainly MSM) 100 mg oral doxycycline twice daily for 7 days (preferred treatment) or One-time 1 g oral dose of azithromycin	N/A
Chronic ReA [75,91,92]		NSAIDS (the standard treatment of choice in severe symptoms) Antibiotic therapy for three to six months (Doxycycline 2 × 100 mg together with rifampicin 1 × 300 mg for 6 months) DMARDs, e.g., sulphasalazine	Azithromycin 1 × 500 mg for 5 days, then 500 mg every 2 weeks, together with rifampicin 1 × 300 mg for 6 months
Ophthalmia neonatorum or chlamydial neonatal pneumonia [83,85,127]		Erythromycin base or ethyl succinate 50 mg/kg body weight/day orally, divided into 4 doses daily for 14 days or Azithromycin 20 mg/kg/day for 3 days	Adverse effect of erythromycin or azithromycin: IHPS or intestinal obstruction among infants aged <6 weeks
Conjunctivitis in adults [128,129]		Doxycycline 2 × 100 mg for 1–3 weeks	Erythromycin 4 × 250 mg for 1–3 weeks
Trachoma [83,130]		Azithromycin (starting at 20 mg/kg up to 1 g) p.o, (single dose)	Erythromycin 500 mg p.o. twice daily for 14 days or Doxycycline 100 mg p.o. for 10 days 1% tetracycline ointment topically Severe stage (trichiasis) → surgery
Lymphogranuloma Venereum [11,76,84,131]		Doxycycline 100 mg orally twice daily for 21 days	Azithromycin 1 g orally once weekly for 3 weeks (NAAT after 4 weeks of a regiment should be carried out to confirm bacterial eradication because the regimen with azithromycin has not been validated) or Erythromycin base 500 mg orally 4 times/day for 21 days Additional procedures: Aspiration and drainage of the ulcers Empirical treatment of sexual partners (100 mg oral doxycycline twice daily for 7 days or a one-time dose of 1 g oral azithromycin)

p.o.—per os, orally; MSM—men who have sex with men; ReA—reactive arthritis; NSAIDs—non-steroidal anti-inflammatory drugs; DMARDs—disease-modifying anti-rheumatic drugs; IHPS—infantile hypertrophic pyloric stenosis.

**Table 3 microorganisms-11-01283-t003:** The mechanisms of antibiotic resistance in *C. trachomatis* infection.

Antibiotic Resistance in *C. trachomatis* Infection			
Antibiotics	Mechanism of Action	Mechanism of Antibiotic Resistance/Available Data	References
Tetracyclines (TET)	Block bacterial protein synthesis by preventing aminoacyl tRNAs from interacting with ribosomes	Genes encoding TET efflux pumps Ribosomal protection proteins Host enzyme inactivation	[139]
		Presence of foreign genomic islands (ranging in size from 6 to 13.5 kb) that integrates into the chlamydial chromosome	[140]
Rifamycins (RIF)	Interact with the β-subunit of RNA polymerase to inhibit bacterial transcription	Resistance to *C. trachomatis* after exposure to subinhibitory concentrations of drug in vitro	[141]
		Resistance development through nucleotide changes in the RNAP β-subunit gene, *rpoB*	[142]
		Strain TW-183 develops resistance to rifalazil when passaged in subinhibitory concentrations of the drug Acquired mutations in *rpoB*	[143]
Fluoroquinolones	Inhibit DNA gyrase and DNA topoisomerase IV	*C. trachomatis* can develop quinolone resistance in vitro when exposed to subinhibitory concentrations of the antibiotic	[144]
		Quinolone-resistant strains are characterized by a point mutation in the quinolone-resistance-determining region of *gyrA*	[145]
Aminoglycosides	Interfere with translation initiation by interacting with the 30S ribosome	Resistant strains carry mutations in the 16S rRNA gene at the KSM binding site (KSM—kasugamycin; antibiotics used to generate aminoglycoside-resistant chlamydial strains)	[146]
		Spectinomycin-resistant *C. trachomatis* L2 mutants encode two nearly identical copies of rRNA operons and drug target sites	[146]
Sulfonamide and trimethoprim (SFM-TMT)	Interferes with bacterial folate synthesis, which is critical for DNA synthesis, repair and methylation	Stable trimethoprim-resistant mutants are infrequent in cultured *C. trachomatis* in vitro in subinhibitory concentrations of the antibiotic	[147]
		Specific insertions, repeats, and point mutations in the *folP* gene (dihydropteroate synthase) confer stable resistance to sulfa drugs Mutations in the *folA* gene (dihydrofolate reductase) confer resistance to trimethoprim	[148]
Azithromycin (a front-line drug for the treatment of chlamydia infections)	Macrolide, which causes bacterial protein synthesis inhibition	*C. trachomatis* L2 strain was selected in lower concentrations of AZM	[146]
		AZM-resistant strains were isolated after exposure to inhibitory concentrations of AZM, while the modestly resistant (AZM tolerant) *C. trachomatis* strain was isolated only after exposure to subinhibitory concentrations of the antibiotic	[138]
		The AZM-tolerant *C. trachomatis* strain harbored a mutation in *rplD* that encodes the ribosomal protein L4	[149]
		The drug-tolerant *C. trachomatis* strain did not grow well in the absence of antibiotics, formed smaller plaques, and produced fewer infectious particles than wild-type parent strains	[146]
Lincomycin	Lincosamide; a bacteriostatic protein synthesis inhibitor, which causes premature dissociation of peptidyl-tRNA from the ribosome	The resistant mutants carried mutations in both 23S rRNA genes	[147]

TET—tetracyclines; RNA—ribonucleic acid; tRNA—transfer RNA; RIF—rifamycins; RNAP—RNA polymerase; DNA—deoxyribonucleic acid; rRNA—ribosomal RNA; KSM—kasugamycin; SFM-TMT—sulfonamide trimethoprim; AZM—azithromycin.

## Data Availability

No new data were created.

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
