# Peer review of "Chlamydia trachomatis—An Emerging Old Entity?"

_microorganisms, 2023, doi:10.3390/microorganisms11051283_

Round 1

Reviewer 1 Report

This review aims to summarize global Chlamydia trachomatis infection. The review topic is conciseness, the content of the review is significant.

Chlamydia trachomatis infection is a global health issue, especially in developing countries. Halt the spread and transmission of Chlamydia trachomatis infection will benefit a large population of people of the world.

There are minor revisions may need:

1.    There are several sentences you need to rewrite. Because you just change several words from the original literature.

2.    Line 102:  C. trachomatis should be italicized.

Author Response

Dear Reviewer,

Thank you for your revision and remarkable suggestions. We implemented adequate corrections according to your suggestions.

Bogna Grygiel-Górniak and Barbara Folga

Reviewer 2 Report

This review has described the prevalence of Chlamydia trachomatis infections and suggests ways to combat the spread of such infections. New ideas for preventing a worldwide epidemic of C. trachomatis are presented in this article, but some problems remain.

1. The diagrams in section 3.3 can be supplemented with pathogens from various transmission routes to make the whole process clearer.

2. In chapter 3.4, is the classification of clinical symptoms based on gender? The text shows in several places that the infection varies among people of different ages, so why not according to age? It would be better to have some research to explain.

3. Most clinical complications are the result of multiple pathogen interactions in which C. trachomatis does not seem to have played a role in altering the outcome of the infection. Therefore, some cases where C. trachomatis caused complications and played a dominant role need to be screened for illustration. On the other hand, the influence of the order of infection of the host by the pathogen on the outcome of the infection can also be presented.

4. In chapter 3.7, the presence of relevant antibodies to detect C. trachomatis does not appear to be specific. The human body produces lgG antibodies in response to stimulation by numerous pathogenic bacteria, such as Staphylococcus aureus, so is it feasible to use lgG antibody detection?

5. “Chlamydia trachomatis” first occurrence should be with full name and all subsequent occurrences with the abbreviation.

Author Response

(The authors gave the same response as above.)

Reviewer 3 Report

Comments to authors:

-The current study is interesting; however, the authors should address the following comments to improve the quality of the manuscript:

-The manuscript should be revised for English editing and grammar mistakes.

- Please write the scientific names of bacterial pathogens in the correct form all over the manuscript and the references section.

Title:

I think the work would benefit from the title that contains the main conclusion of the study (should be derived from the conclusion). Please modify the title.

Abstract:

- The abstract must illustrate the used methods and the most prevalent results (give more hints about methods and results). Besides, rephrase the aim of the work and the main conclusion of your findings.

-A graphical abstract is recommended (If possible).

- Add the full expression before the abbreviations.

-Introduction: (it needs to be more informative):

-Give a hint about the virulence factors and the mechanism of disease occurrence , and infecions caused by Chlamydia trachomatis .

- The authors should illustrate the public health importance concerning the emergence of multidrug-resistant (MDR) bacterial pathogens that reflect the necessity of new potent and safe antimicrobial agents. Several studies proved the widespread MDR- bacterial pathogens;

Authors could add the following paragraph:

Multidrug resistance has been increased all over the world that is considered a public health threat. Several recent investigations reported the emergence of multidrug-resistant bacterial pathogens from different origins that increase the necessity of the proper use of antibiotics. Besides, the routine application of the antimicrobial susceptibility testing to detect the antibiotic of choice as well as the screening of the emerging MDR strains. You are advised to cite the following valuable studies:

1. PMID: 20860486

2. PMID: 36365013

3. PMID: 36818807

- Illustrate the mechanism of action different virulence factors of Chlamydia trachomatis.

-Rephrase the aim of the work to be clear and better sound.

Material and methods:

- Support all methods with updated specific references.

  -Results:

- Please add a starting paragraph to the results section to briefly introduce the topic, your goals and 

hypothesis and a short summary of what you did in this work.

-Add this subtitle: Genomic features beyond Chlamydia trachomatis phenotypes: explain in detail.

- Please illustrate the Complications associated with the treatment of chlamydial infections.

- Please illustrate Heterotypic resistance in Chlamydiae

- Please illustrate Chlamydial resistance to individual antibiotic classes.

-Please illustrate Utility of antibiotic resistance in chlamydial genetics, recombination & transformation.

-Increase the resolution of all figures (must be 600 dpi).

-Discussion:

 The authors are advised to illustrate the real impact of their findings without repetition of results.

Please illustrate different mechanisms of antibacterial resistance in Chlamydia.

-Conclusion

- Should be rephrased to be sounded. A real conclusion should focus on the question or claim you articulated in your study, which resolution has been the main objective of your paper?

Please note that the Author contributions section is missed.

Author Response

(The authors gave the same response as above.)

Round 2

Reviewer 3 Report

The authors have carried out significant changes to the manuscript. They have addressed most of the suggested corrections and comments. Really, it's an interesting study that has a significant impact. Now, the manuscript could be accepted.

Congratulations.

Author Response

Dear Editor,
We have considered all your suggestions and inserted all the references and citations within the text.
We provide an upside-down list of the steps depicted in Figure 3.
Kind regards
Bogna Grygiel-Górniak
